# Molecular determinants of substrate specificity in the efflux pump CraA from *Acinetobacter baumannii*

Wuen Ee Foong,[1,2,3] Xinxin Xiang,[1] Klaas M. Pos,[3] Heng-Keat Tam[1,2,3,4]

**ABSTRACT** The major facilitator superfamily (MFS) type efflux pumps of *Acinetobacter baumannii* play important roles in antibiotic resistance. However, the molecular mechanism of these transporters remains poorly understood. To address the molecular basis of substrate polyspecificity mediated by multidrug MFS transporters, we compared the substrate binding modes of *A. baumannii* CraA with its well-studied homolog, *Escherichia coli* MdfA. MdfA and CraA share similar structural features, including a cavity accessible to drugs from the cytoplasm when these transporters adopt the inside-out conformation. This predominantly hydrophobic cavity contains several distinct titratable and hydrophilic residues. Through substitution analysis, we demonstrate that these polar residues within the CraA drug binding cavity contribute to the transport of all tested drugs, whereas mutations of hydrophobic residues result in altered drug recognition profiles. In addition to the known titratable residues E38 and D46, we identified E338 as the only titratable residue that plays a substrate-specific role, as it is required for efficient transport of norfloxacin, but not ethidium. Substitution of E338 with asparagine or glutamine changes substrate specificity, enabling specific recognition of phenicols and mitomycin C. Furthermore, we show that the aromaticity of Y42 is crucial for phenicol recognition, while general hydrophobicity at this position is critical for mitomycin C specificity. We propose that E338 and Y42 function as key substrate selectivity determinants in CraA.

**IMPORTANCE** Multidrug efflux transporters of the major facilitator superfamily (MFS) are key contributors to antibiotic resistance, mediating the export of structurally diverse compounds across bacterial membranes. While homologous transporters such as *Escherichia coli* MdfA and *Acinetobacter baumannii* CraA share high structural similarity and overlapping substrate profiles, the molecular basis of their substrate specificity remains poorly understood. In this study, we show that structural homology among MFS transporters does not inherently imply mechanistic conservation, as species-specific variations can give rise to distinct substrate recognition profiles. Our findings reveal that CraA utilizes unique residues Y42 and E338 for substrate selectivity, while R124 and Y73 contribute to its transport activity. These findings enhance our understanding of efflux pump specificity and underscore the need to consider organism-specific features when targeting multidrug transporters in antimicrobial therapy.

**KEYWORDS** *Acinetobacter baumannii*, multidrug resistance, efflux pumps, major facilitator superfamily, substrate polyspecificity

*A*cinetobacter baumannii is a worldwide opportunistic pathogen responsible for nosocomial infections, which are high in mortality rate, such as ventilator-associated pneumonia, bloodstream infection, urinary tract infection, meningitis, and wound infection (1). Therefore, carbapenem-resistant *A. baumannii* is repeatedly classified among the WHO priority pathogens list, requiring the "critical" urgency in need for

**Peer Reviewer** Lan Guan, Texas Tech University Health Sciences Center, Lubbock, Texas, USA

Address correspondence to Klaas M. Pos, pos@em.uni-frankfurt.de, or Heng-Keat Tam, tamhk60@hotmail.com.

The authors declare no conflict of interest.

See the funding table on p. 15.

antibiotic research (2). *A. baumannii* is notoriously difficult to eradicate due to its high level of intrinsic antibiotic resistance and its capability in the acquisition of multidrug resistance genes (3, 4). Among the different classes of antibiotic resistance, multidrug efflux pumps are one of the most prominent contributors to multidrug resistance in *A. baumannii*, in addition to the outer membrane impermeability (5, 6). Of particular interest, *A. baumannii* intrinsically possesses a large repertoire of multi-drug efflux pumps classified into six different superfamilies: the ATP binding cassette, multidrug and toxic compound extrusion, resistance nodulation cell division (RND), small multidrug resistance, proteobacterial antimicrobial compound efflux, and major facilitator superfamily (MFS) (6).

To date, the functional relevance and molecular determinants of *A. baumannii* MFS transporters in antibiotic resistance have not been addressed as intensively as the RND transporters, like the AdeABC and AdeIJK systems. The MFS is the most diverse and largest secondary active transporters known to date (7). The tetracycline efflux pumps such as Tet(A) and Tet(B) are widely present in *A. baumannii* clinical isolates, and these efflux pumps are reported to be ineffective against glycylcyclines (8, 9). Recently, it was reported that TetA of *A. baumannii* AYE confers transport activity against tigecycline in synergy with the RND efflux pumps, suggesting an unexplored functional redundancy of MFS and RND in the removal of tigecycline (10). Similar to the well-studied *Escherichia coli* MdfA drug/H$^+$ antiporter, the *A. baumannii* MFS transporters CraA, ABAYE0913, and AmvA display a broad spectrum of drug specificity (11, 12). Other characterized *A. baumannii* MFS efflux pumps, like AbaF and CmlA5, specifically transport fosfomycin and phenicols, respectively (12, 13), and AbaQ is specific for quinolone transport (14).

The MFS efflux pump CraA is a homolog of *Escherichia coli* MdfA, which is widely present in clinical isolates of *A. baumannii* (15, 16). Inactivation of *craA* in *A. baumannii* ATCC 19606 resulted in a 128-fold decrease in the MIC value of chloramphenicol, suggesting it is a specific chloramphenicol transporter (15). However, our previous study has shown that when CraA was overproduced in *E. coli*, it transports a plethora of structurally dissimilar compounds. These include electroneutral drugs, mono- or divalent cations, and zwitterionic compounds, such as chloramphenicol, resembling the multidrug specificity shown for MdfA (12, 17). For MdfA, Glu-26, Asp-34, and Arg-112 are the most evolutionarily conserved membrane-embedded titratable residues (18, 19). These residues can also be found in CraA at homologous positions, i.e., Glu-38, Asp-46, and Arg-124 (Fig. S1). The drug/H$^+$ antiport mechanism of MdfA is postulated to involve a competition between proton and substrate binding to a substrate binding cavity involving Glu-26 and Asp-34 (20, 21). Yet, neither Glu-26 nor Asp-34 is essential for drug binding. Remarkably, Glu and/or Asp can be placed somewhere else inside the large binding pocket without major effect on drug efflux. These observations suggest that the coupling between substrate binding and protonation/deprotonation is highly flexible and dynamic (18, 20, 22–25). Apart from Glu-26 and Asp-34, Arg-112 plays an essential role in the transport activity of MdfA. Alanine substitution of Arg-112 renders the cells sensitive to all substrates, apart from the MdfA_R112H variant, which retains partial activity against cationic drugs, phenicols, and fluoroquinolones (19). In the MdfA homolog CraA, Asp-46 appears essential for binding of cationic compounds, but not for H$^+$ binding, while Glu-38 is involved in the substrate binding and/or proton translocation (12). Interestingly, the drug binding pocket of CraA differs from MdfA as it contains another titratable residue, Glu-338, which is involved in substrate recognition (12).

Here, we describe a detailed phenotypic characterization of the drug susceptibility profiles of 30 alanine substitution variants of CraA. The targeted residues line the substrate binding pocket. The variants are characterized by overexpression of the *craA* mutant genes in *E. coli* BW25113 Δ*emrE*Δ*mdfA*. Our study was guided by building a CraA model based on the structure factor-derived electron densities of MdfA (26) and by *in silico* prediction of binding modes of four drugs with distinct physicochemical properties. These drugs, chloramphenicol, ethidium, mitomycin C, and norfloxacin, were selected on the basis of their activity against an *A. baumannii craA* knockout strain. Our data

suggest that the strong homology and overlapping substrate specificities between CraA and MdfA do not lead to a similar molecular transport mechanism.

## MATERIALS AND METHODS

### Cloning of the *craA* gene and site-directed mutagenesis

The *craA* gene was cloned into the pTTQ18 vector by Gibson Assembly as previously described (12). Amino acid substitutions were introduced using the ExSite site-directed mutagenesis protocol (Stratagene). The inserted genes were verified by sequencing (Eurofins). All cloning primers are listed in Table S1.

### Drug agar plate assay in *E. coli*

Drug susceptibility assays were performed as previously described (12). Briefly, cell cultures of *E. coli* BW25113 Δ*emrE*Δ*mdfA* harboring empty vector (pTTQ18), pTTQ18_CraA_WT, and *craA* mutants were diluted to $OD_{600}$ $10^0$–$10^{-5}$ and 4 µL of the diluted cultures were spotted on LB agar plate supplemented with 100 mg/L ampicillin or 50 mg/L carbenicillin, 0.2 mM isopropyl-β-D-thiogalactoside (IPTG), and tested drugs (Fig. S2 and S3). Plates were incubated at 37°C overnight. The expression of *craA* and *craA* mutants was determined by detecting the proteins produced by western blot analysis using anti-His-alkaline-phosphatase antibody (Sigma-Aldrich). Cell growth was scored based on the number of dilution steps that exhibited cell growth, with the highest score of 6 representing growth to the last dilution, while 0 represents no growth. To calculate the relative cell growth, the cell growth of each variant was normalized with the cell growth of wild-type (WT) CraA on the same drug agar plates. The *P* value was calculated by a two-sided Student's *t*-test.

### Construction of the *craA* gene knock-out and complementation

The *craA* gene of *A. baumannii* ATCC19606 was inactivated as previously described (12). All primers used are listed in Table S1. Approximately 1,500 bp of up- and downstream regions of *craA* were amplified from the genome of *A. baumannii* ATCC 19606 and inserted into pBIISK_sacB/kanR using the PstI, BamHI, and NotI restriction sites. The constructed plasmid was transformed into *A. baumannii* by electroporation (2.5 kV, 200 'Ω and 25 µF) and plated onto LB agar supplemented with 50 mg/L kanamycin. Counter-selection was performed with LB media supplemented with 10% sucrose. A complementation plasmid containing the *craA* gene and the 700 bp upstream region of *craA* was inserted into pBAV1K as previously described (12).

### Ethidium efflux assays and statistical analysis

Ethidium efflux assays were performed as previously described (12). *E. coli* BW25113 Δ*emrE*Δ*mdfA* harboring pTTQ18, pTTQ18-CraA_WT, or *craA* mutants were induced with 0.2 mM IPTG for 2 h, harvested, washed twice with Dulbecco's phosphate-buffered saline (DPBS: 8 g/L NaCl, 0.2 g/L KCl, 1.44 g/L $Na_2HPO_4.2H_2O$, 0.2 g/L $KH_2PO_4$, 10 mM $MgCl_2$), and resuspended in the same buffer. To abolish the proton gradient, induced cells were incubated with 40 µM of carbonyl cyanide *m*-chlorophenyl hydrazone (CCCP) at 37°C for 10 min and further incubated with 10 µM ethidium bromide for 45 min. After incubation, cells were washed with DPBS and subsequently resuspended in the same buffer. Ethidium transport was initiated by adding 0.36% glucose to the cell suspension, and the fluorescence was measured using the Tecan Microplate Reader (Tecan, Switzerland) (excitation and emission wavelengths of 535 and 610 nm, respectively). To evaluate the effect of CCCP on ethidium efflux, CCCP (40 µM) was added to the cell suspension, and fluorescence was monitored for 1 min. Subsequently, glucose (0.36%) was added, and fluorescence measurement was continued for an additional 8 min.

Each of the ethidium efflux curves following glucose addition was individually analyzed by non-linear least squares regression with the Levenberg-Marquardt algorithm

using multiple starting values ("nls.multstart" package) in R 4.4.1 (27, 28) and fitted to the sigmoid function (equation 1), with fixed upper asymptote (at y = 1).

$$y(t, H, S, Y_0) = 1 - \frac{1 - Y_0}{1 + \exp\left(\frac{H - \log t}{S}\right)},\tag{1}$$

where $H$ is the log of the time required for the cells to remove half of the ethidium (equivalent to the relative fluorescence of ethidium of ~0.5), $S$ determines the steepness of the curve, and $Y_0$ is the lower asymptote.

The estimation of the parameter $H$ of each CraA variant is highly correlated with the wild-type CraA and cells harboring an empty vector within an experiment (i.e., technical replicates) as well as between different biological replicates. Therefore, the parameter $H$ of each data set (i.e., all technical replicates within biological replicates) was analyzed by a linear mixed effects model by maximum likelihood with a random intercept for each replicate within experiments using equation 2 executed via the "lme4" package in R 4.4.1 (27, 29). Pairwise comparison of each of the CraA variants was analyzed by general linear hypothesis testing and Tukey's multiple comparison test using the "multcomp" package in R 4.4.1 (27, 30).

$$H \sim 0 + \text{Genotype} + (1 \mid \text{Exp/Rep}),\tag{2}$$

where $H$ is the log of the time until the relative fluorescence is ~0.5 (obtained from equation 1), genotype represents each of the CraA variants, Exp represents biological experiments, and Rep is technical replicates within a biological experiment.

## Norfloxacin accumulation assay in whole cells

Norfloxacin accumulation assays were performed as previously described (31) with slight modification. *E. coli* BW25113 Δ*emrE*Δ*mdfA* harboring pTTQ18, pTTQ18-CraA_WT, or *craA* mutants were induced with 0.2 mM IPTG for 2 h, harvested, washed twice with DPBS, and resuspended in the same buffer to an $OD_{600}$ of 1.0. Cells were incubated with 100 µg/mL norfloxacin for 5 min at room temperature, harvested, and washed twice with DPBS. Subsequently, cells were resuspended with 800 µL Lysis Buffer (1% SDS in 0.1 M glycine HCl, pH 3) for 10 min at room temperature. The fluorescence was measured using the Tecan Microplate Reader (excitation and emission wavelengths of 281 and 440 nm, respectively).

The fluorescence intensity of each CraA variant was normalized with the fluorescence intensity of wild-type CraA set to 1. Subsequently, the normalized data sets (all technical replicates within biological replicates) were analyzed by a linear mixed effects model by restricted maximum likelihood with a random intercept for each replicate within experiments using equation 3 (executed via the "lme4" package in R 4.4.1 (27, 29). Pairwise comparison of each of the CraA variants was analyzed by general linear hypotheses testing and Tukey's multiple comparison test using the "multcomp" package in R version 4.3.3 (27, 30).

$$\text{Fluorescence} \sim 0 + \text{Genotype} + (1 \mid \text{Exp/Rep}),\tag{3}$$

where "Fluorescence" is the normalized fluorescence intensity measurements, Genotype represents each of the CraA variants, Exp represents biological experiments, and Rep is technical replicates within a biological experiment.

## Reverse transcription qPCR

To analyze the gene expression of *craA*, overnight cultures of *A. baumannii* ATCC19606 strain were inoculated into 50 mL of LB medium supplemented with 8 µg mL$^{-1}$ of chloramphenicol, to an initial $OD_{600}$ of 0.05. Cell cultures were subsequently incubated at 37°C with 130 rpm to an $OD_{600}$ of 0.5 to 0.7. Cells were also grown in LB medium

without chloramphenicol as a control. Cell suspensions (1 mL of $OD_{600} = 1.0$) were collected and treated with RNAprotect bacteria reagent (Qiagen). RNA extraction was conducted using the RNeasy Mini Kit (Qiagen), and DNA was removed by on-column DNA digestion with Turbo DNase (Invitrogen). RNA samples were subsequently purified by the RNeasy MinElute Cleanup kit (Qiagen). One-step reverse transcription-quantitative polymerase chain reaction (RT-qPCR) was performed on a thermal cycler Rotor-Gene Q (Qiagen) using QuantiNova SYBR Green RT-PCR kit (Qiagen). The primers used were listed in Table S1, with *rpoB* as the reference gene. Gene expression was analyzed as previously described (10). The $\Delta C_T$ values were calculated as $C_{T(rpoB)} - C_{T(efflux\ pump)}$. Mean differences in $\Delta C_T$ values between the "Treatment" group and "Control" group were analyzed by an analysis of variance model in R version 4.3.3 (27). $\Delta\Delta C_T$ values represent the mean difference in $\Delta C_T$ between samples in the presence or absence of antibiotic treatment.

## CraA homology model

Since CraA shares significantly high sequence similarity to MdfA (44% sequence identity, 64% similarity (Fig. S1), the structure factors from our published data set of *Escherichia coli* MdfA Q131R_L339E mutant (PDB: 6EUQ) (26) were extracted, and the electron density of this structure was used as a template to build a CraA homology model. We assume that the overall resolution of a diffraction data set of 2.2 Å of reflections will provide a sufficient and informative electron density map to fit the backbone of amino acids as well as the side chain of the conserved residues into the electron density map of 6EUQ to refine the model and map until convergence (26). Subsequently, the model of the CraA putative binding pocket was built by fitting the side chain of the non-conserved residues into the map with the highest rotamer probability, unless the orientation of these rotamers caused steric clashes with the nearby residues, and refining the model and map until convergence. The tunnel of CraA homology model was calculated by CAVER 3.0 (32).

## Molecular docking

Molecular docking calculations were performed using the software AutoDock Vina (33). Protein and ligand input files (chloramphenicol, ethidium, mitomycin C, and norfloxacin) were prepared with AutoDock Tools (34). Flexibility of the residues (E38, Y42, N45, D46, M70, Y73, L74, L131, S135, S162, L163, L246, M247, I250, F276, L279, N283, E338, F342, L368, M369, and F372) lining the cavity of the protein was taken into account for the molecular docking calculations, as the substitution of these residues revealed a distinct from wild-type CraA drug susceptibility profile. For the first docking calculations, the search of poses was performed within a large cubic volume of $46 \times 38 \times 30$ Å$^3$ covering the whole cavity of the protein, with an exhaustiveness of 20 (default 8). After identification of putative binding pockets within the cavity of the protein, the search for docking poses was performed within a cubic volume of $24 \times 24 \times 24$ Å$^3$ and centered at the cavity of the protein (x = 28.662 Å; y = 3.413 Å; z = 23.172 Å), with an exhaustiveness of 120.

## RESULTS

### Phenotypic analysis of *A. baumannii* 19606 Δ*craA* and complementation

Previously, we showed that expression of *craA* in *E. coli* Δ*emrE*Δ*mdfA* reduced the susceptibility for phenicols, ethidium, norfloxacin, and mitomycin C (12). We sought to address whether CraA confers resistance against these substrates in its native host, *A. baumannii*. We therefore deleted *craA* from the chromosome of *A. baumannii* ATCC19606 by homologous recombination and determined its drug resistance profile. *A. baumannii* Δ*craA* was shown to be highly sensitive toward the above-mentioned drugs, and complementation of this strain by the plasmid pBAV1K-*craA* rescued wild-type suscepti-

bility (Fig. S4). The results confirm that CraA is a broad-spectrum multidrug transporter and displays this phenotype also in *A. baumannii*.

## Molecular docking of substrates in CraA

The homology model of CraA aligned well with the AlphaFold2-predicted structure (35), with 2.34 Å r.m.s.d. of $C_\alpha$ over 400 residues (Fig. 1A), supporting its reliability for downstream molecular docking studies. To validate our docking protocol, we used chloramphenicol as a model ligand, leveraging the availability of the MdfA-chloramphenicol crystal structure (PDB: 4ZOW) as a reference (36). Superimposition of the CraA-chloramphenicol docking model with the MdfA-chloramphenicol crystal structure demonstrated a comparable binding pose (Fig. 2A, 3A and B; Fig. S5A and B), thereby validating the docking approach and reinforcing the reliability of our homology model in predicting substrate interactions. Subsequent docking experiments with three additional CraA substrates, namely, ethidium, norfloxacin, and mitomycin C, revealed binding of these substrates to the apex of the CraA binding cavity proximal to the periplasmic side of the inner membrane (Fig. 2B). Like the chloramphenicol docking pose (Fig. 3B; Fig. S5B), all these substrates engaged primarily through hydrophobic or van der Waals interactions (Fig. 3C through E; Fig. S5C through E). Interestingly, ethidium, mitomycin C, and norfloxacin shared overlapping binding modes involving π-interaction between Y42 and the aromatic moieties of these ligands (Fig. 3C through E; Fig. S5C and E). In contrast, chloramphenicol adopted a distinct binding pose compared to that of the former drugs (Fig. 3B; Fig. S5B).

## Functional importance of membrane-embedded titratable residues

E38 and D46 reside at the N-terminal half of CraA and are accessible from the cytoplasm in the inward open conformation (Fig. 1B). Alanine substitution at these positions abolished resistance to all tested drugs, underscoring their functional importance in protonation and/or substrate recognition (12, 24). Notably, the CraA double substitution

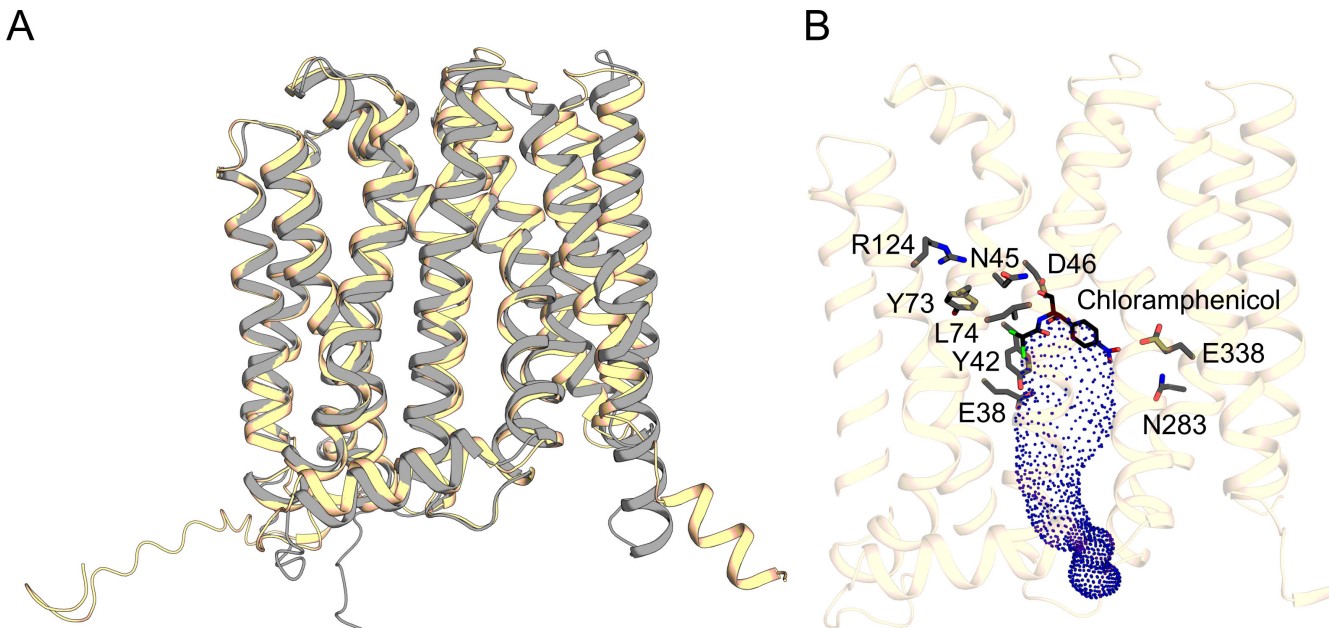

**FIG 1** Homology model of CraA and key residues in the substrate binding pocket. (A) The homology model of CraA was calculated based on the structure factors of the MdfA_Q131R_L339E variant (PDB: 6EUQ; 26). The cartoon representations of the CraA homology model and the AlphaFold2-predicted structure (35) are colored beige and gray, respectively. (B) The inward-facing cavity was calculated using CAVER 3.0 (32) and is shown as a dot-surface representation. Key residues that are important for drug binding, transport, and protein integrity are depicted as sticks. Chloramphenicol, obtained from the crystal structure of *E. coli* MdfA (PDB: 4ZOW; 36), is overlaid onto the CraA homology model, showing that chloramphenicol is bound at the apex of the CraA binding pocket.

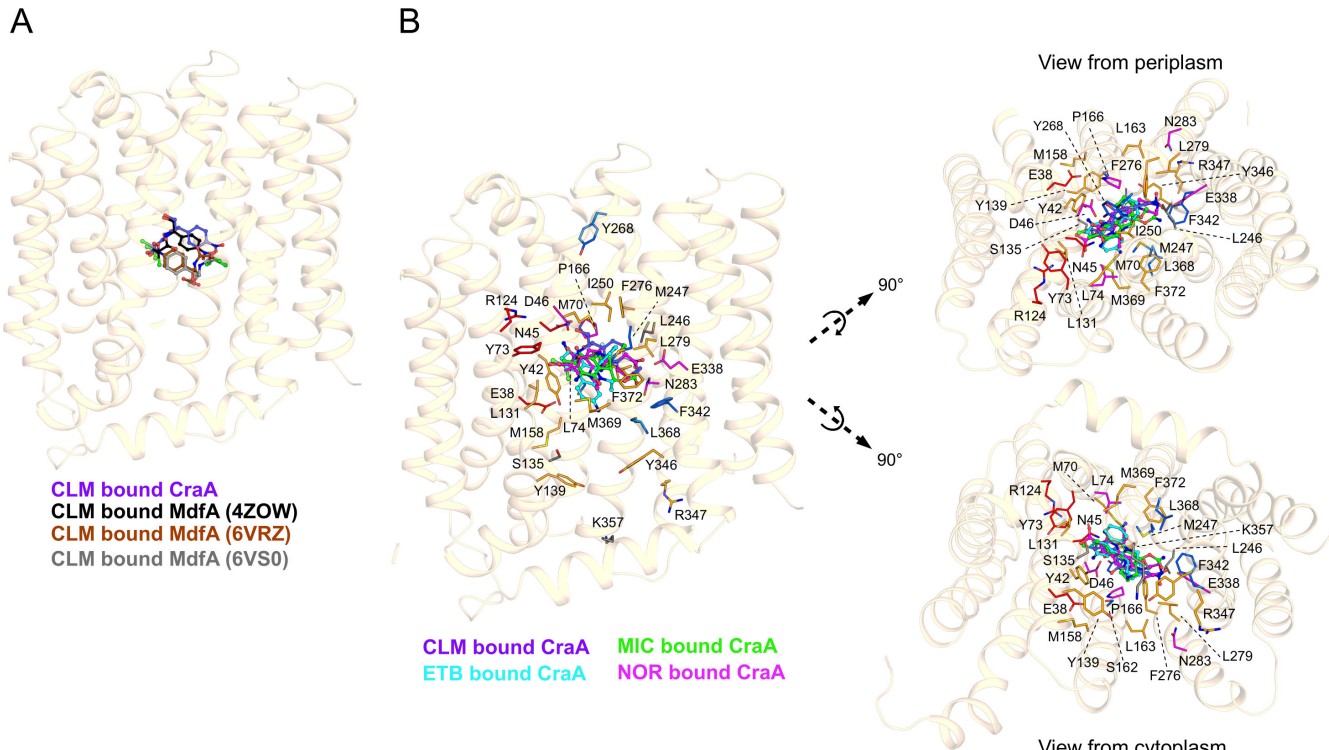

**FIG 2** *In silico* modeling of chloramphenicol, ethidium, mitomycin C, and norfloxacin binding modes. The CraA homology model was created based on the MdfA-Q131A_L339A variant (PDB: 6EUQ) (26). (A) Chloramphenicol, CLM (carbon: gray/black/purple/brown; nitrogen: blue; oxygen: red; chlorine: green) docking solution for CraA homology model (purple) agrees well with the binding mode of chloramphenicol to *E. coli* MdfA (PDB: 4ZOW; black) (36), but does not agree well with the binding mode of chloramphenicol to *E. coli* MdfA_E26T-D34M-A150E (PDB: 6VRZ, 6VS0) (25). (B) Substrate docking solutions in the CraA cavity indicate that the tested drugs (CLM: chloramphenicol; ETB: ethidium; MIC: mitomycin C; NOR: norfloxacin) bind to the apex of the CraA binding pocket proximal to the periplasmic side. Alanine-scanning mutagenesis of the residues lining the putative binding pocket (residues shown in red: severely or mildly affected against all drugs; residues shown in magenta: severely or mildly affected against >6 drugs; residues shown in orange: severely or mildly affected against 3–5 drugs; residues shown in marine: severely or mildly affected against 1–2 drugs; residues shown in gray: unaffected).

variants E38A_D46A, E38D_D46A, and E38A_D46E were inactive against all tested drugs, while the E38D_D46E variant retained wild-type activity specifically against chlorhexidine (Fig. 4; Fig. S3A).

In *E. coli* MdfA, the corresponding E26 (E38 in CraA) and D34 (D46 in CraA) are individually dispensable for drug binding (23, 24, 37). Substituting E26 with glutamine preserved resistance toward neutral drugs but abrogated resistance toward cationic drugs (18). The CraA_E38Q variant, however, conferred partial resistance to monocationic benzalkonium and the neutral phenicols, but not against norfloxacin nor monocationic ethidium (Fig. 4; Fig. S3A). As anticipated, alanine substitution of CraA_D46 altered the drug resistance profile (Fig. 4; Fig. S3A) (12). While the D46A variant retained wild-type resistance to norfloxacin and florfenicol, substitution with histidine (D46H) abrogated resistance to both norfloxacin and phenicols (Fig. 4; Fig. S3A). Interestingly, replacing D46 with glutamate (D46E) restored resistance to cationic drugs and conferred partial activity against florfenicol (Fig. 4; Fig. S3A), underscoring the importance of a carboxylate group at this position for the transport of cationic drugs. Intriguingly, the D46N variant displayed a unique multidrug resistance phenotype, maintaining wild-type resistance against florfenicol, thiamphenicol, benzalkonium, and even displaying hyperactivity against norfloxacin (Fig. 4; Fig. S3A). These results suggest a conserved role of E38 and D46 in drug transport.

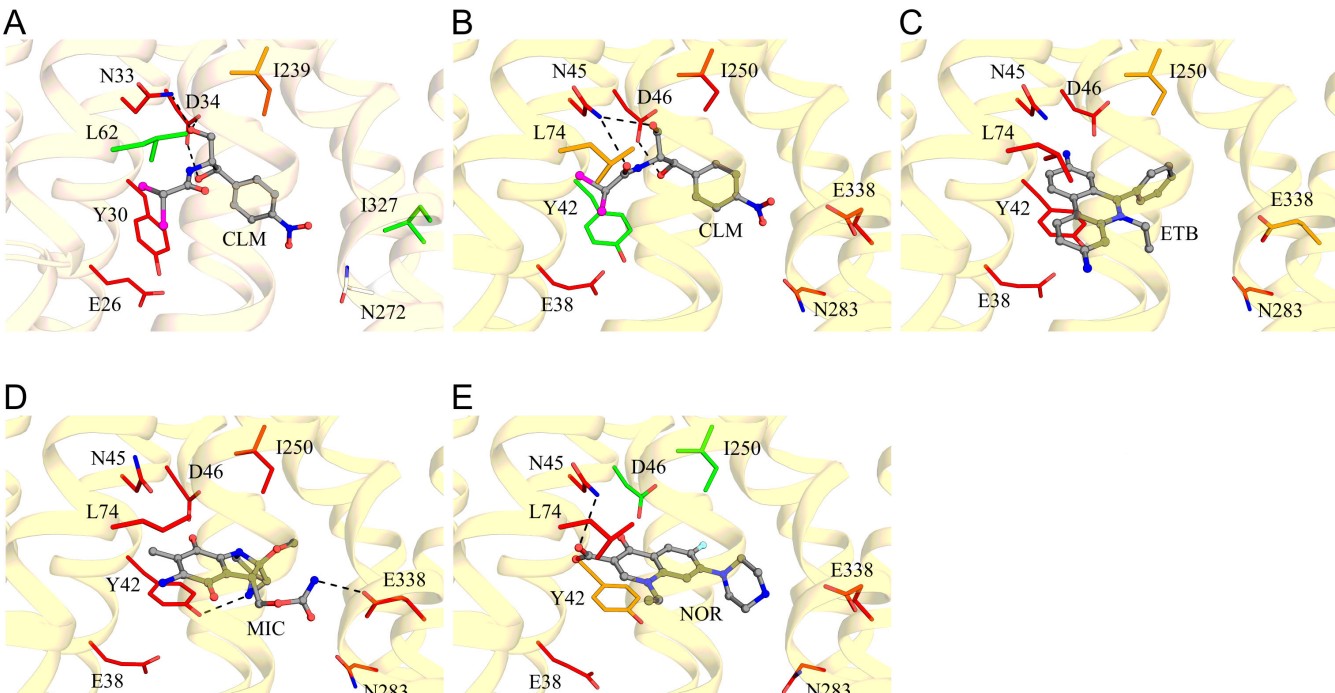

**FIG 3** Substrate modes of MdfA and CraA. (A) Co-crystal structure of MdfA with chloramphenicol (carbon: white/gray; nitrogen: blue; oxygen: red; chlorine: magenta). Substrate docking solutions for (B) chloramphenicol (CLM); (C) ethidium (ETB); (D) mitomycin C (MIC); and (E) norfloxacin (NOR), in the CraA binding pocket (carbon: red/orange/green; nitrogen: blue; oxygen: red; chlorine: magenta; fluorine: cyan; sulfur: yellow). Zoomed-in view of Fig. S5 highlighting only the key residues involved in substrate specificity and protonation for clarity. Residues shown in red (carbon): severely affected; residues shown in orange (carbon): mildly affected; residues shown in green (carbon): unaffected; residues shown in white (carbon): unknown.

## Role of E338 in CraA substrate selectivity

Despite its high structural similarity to MdfA, CraA uniquely possesses an additional carboxylate E338 within its binding cavity, corresponding to I327 in MdfA (Fig. S1). Prior studies demonstrated that introducing an extra carboxylate residue, apart from its native E26 and D34, in the MdfA binding pocket broadens its substrate specificity to include divalent cationic compounds with short linkers, such as methyl viologen (Fig. S2) (38). However, CraA does not support the efflux of methyl viologen or related compounds (Fig. S6), indicating that E338 is insufficient to expand the CraA substrate specificity. In line with previous findings (12), the E338A substitution abrogated resistance to most tested drugs, while retaining wild-type activity against ethidium and benzalkonium (Fig. 4; Fig. S3A). Notably, the E338Q and E338N variants conferred wild-type-like resistance to all drugs except chlorhexidine and norfloxacin, where full susceptibility was observed (Fig. 4; Fig. S3A). In contrast, the E338D variant maintained resistance to benzalkonium, chlorhexidine, and norfloxacin (Fig. 4; Fig. S3A), suggesting that this residue contributes to substrate selectivity rather than general transport function.

## Divergent roles of conserved residues R124 and Y73

R124 in CraA, homologous to R112 in MdfA, is among the most evolutionarily conserved residues in MdfA and is critical for its transport function (Fig. S1) (19). Substitution of R124 to alanine or histidine abolished drug resistance in CraA (Fig. 4; Fig. S3A), contrasting with the partial activity of MdfA_R112H against cationic drugs, phenicols, and fluoroquinolones (19). Structural analyses of MdfA indicate that R112 participates in a hydrogen-bonding network and a cation-π interaction with Y61 and F174 in both the inward- and outward-open states (Fig. S7; 36, 39). Interestingly, the homologous residue Y73 in CraA exhibited distinct phenotypes compared to MdfA (36). In line with

| Variants | CLM 2.5–5 µg/mL | FLR 2.5–5 µg/mL | THM 20–50 µg/mL | ETB 0.2–0.4 mg/mL | BEN 15–30 µg/mL | CLH 2.5–5 µg/mL | NOR 0.1–0.5 µg/mL | MIC 1–2.5 µg/mL |
|---|---|---|---|---|---|---|---|---|
| WT | 1.000±0.000 | 1.000±0.000 | 1.000±0.000 | 1.000±0.000 | 1.000±0.000 | 1.000±0.000 | 1.000±0.000 | 1.000±0.000 |
| E38A | 0.042±0.042 ** | 0.000±0.000 | 0.125±0.080 ** | 0.000±0.000 | 0.000±0.000 | 0.000±0.000 | 0.000±0.000 | 0.000±0.000 |
| E38Q | 0.792±0.686 ** | 0.396±0.151 ** | 1.000±0.000 | 0.000±0.000 | 0.708±0.105 * | 0.000±0.000 | 0.000±0.000 | 0.000±0.000 |
| E38D | 0.063±0.063 ** | 0.000±0.000 | 1.000±0.000 | 0.000±0.000 | 1.000±0.000 | 0.000±0.000 | 0.000±0.000 | 0.000±0.000 |
| E38H | 0.000±0.000 | 0.000±0.000 | 0.167±0.118 ** | 0.000±0.000 | 0.083±0.083 ** | 0.000±0.000 | 0.000±0.000 | 0.000±0.000 |
| D46A | 0.5.00±0.114 ** | 0.958±0.027 | 0.208±0.125 ** | 0.000±0.000 | 0.667±0.118 * | 0.000±0.000 | 1.000±0.000 | 0.000±0.000 |
| D46N | 0.188±0.102 ** | 0.896±0.063 | 1.000±0.000 | 0.000±0.000 | 1.000±0.000 | 0.000±0.000 | 1.656±0.546 | 0.000±0.000 |
| D46E | 0.021±0.021 ** | 0.542±0.108 ** | 0.167±0.118 ** | 1.000±0.000 | 1.000±0.000 | 1.000±0.000 | 0.133±0.133 ** | 0.000±0.000 |
| D46H | 0.167±0.100 ** | 0.229±0.083 ** | 0.167±0.118 ** | 0.000±0.000 | 0.583±0.144 * | 0.000±0.000 | 0.327±0.093 ** | 0.000±0.000 |
| E38A_D46A | 0.000±0.000 | 0.000±0.000 | 0.000±0.000 | 0.000±0.000 | 0.000±0.000 | 0.000±0.000 | 0.000±0.000 | 0.000±0.000 |
| E38D_D46A | 0.042±0.027 ** | 0.125±0.061 ** | 0.000±0.000 | 0.000±0.000 | 0.000±0.000 | 0.000±0.000 | 0.000±0.000 | 0.000±0.000 |
| E38A_D46E | 0.000±0.000 | 0.021±0.021 ** | 0.000±0.000 | 0.000±0.000 | 0.000±0.000 | 0.000±0.000 | 0.000±0.000 | 0.000±0.000 |
| E38D_D46E | 0.000±0.000 | 0.000±0.000 | 0.000±0.000 | 0.000±0.000 | 0.167±0.000 ** | 0.967±0.033 | 0.000±0.000 | 0.000±0.000 |
| N45A | 0.000±0.000 | 0.000±0.000 | 0.000±0.000 | 0.000±0.000 | 0.071±0.050 ** | 0.000±0.000 | 0.000±0.000 | 0.000±0.000 |
| R124A | 0.000±0.000 | 0.000±0.000 | 0.000±0.000 | 0.000±0.000 | 0.000±0.000 | 0.000±0.000 | 0.000±0.000 | 0.000±0.000 |
| R124H | 0.000±0.000 | 0.000±0.000 | 0.000±0.000 | 0.000±0.000 | 0.000±0.000 | 0.000±0.000 | 0.000±0.000 | 0.000±0.000 |
| S135A | 1.000±0.000 | 1.000±0.000 | 1.000±0.000 | 1.000±0.000 | 0.940±0.039 | 1.000±0.000 | 1.25±0.182 | 1.000±0.000 |
| S162A | 1.000±0.000 | 0.967±0.033 | 1.000±0.000 | 0.972±0.028 | 0.940±0.039 | 0.944±0.056 | 0.379±0.110 ** | 1.000±0.000 |
| N283A | 0.305±0.1 ** | 0.000±0.000 | 0.833±0.043 ** | 0.000±0.000 | 0.513±0.125 ** | 0.000±0.000 | 0.000±0.000 | 0.000±0.000 |
| E338A | 0.033±0.033 ** | 0.407±0.161 * | 0.283±0.083 ** | 0.800±0.069 * | 1.000±0.000 | 0.000±0.000 | 0.061±0.041 ** | 0.000±0.000 |
| E338D | 0.033±0.033 ** | 0.000±0.000 | 0.233±0.083 ** | 0.217±0.086 ** | 1.000±0.000 | 0.926±0.029 * | 1.111±0.111 | 0.000±0.000 |
| E338I | 0.050±0.050 ** | 0.460±0.058 ** | 0.167±0.096 ** | 1.000±0.000 | 1.000±0.000 | 0.000±0.000 | 0.230±0.233 ** | 0.050±0.050 ** |
| E338N | 1.000±0.000 | 1.000±0.000 | 1.000±0.000 | 0.967±0.033 | 1.000±0.000 | 0.000±0.000 | 0.230±0.122 ** | 1.022±0.022 |
| E338Q | 1.000±0.000 | 1.040±0.040 | 1.000±0.000 | 0.917±0.037 * | 1.000±0.000 | 0.000±0.000 | 0.167±0.188 ** | 0.967±0.062 |
| R347A | 0.5.00±0.043 ** | 0.778±0.035 ** | 0.333±0.075 ** | 1.000±0.000 | 1.000±0.000 | 1.000±0.000 | 0.500±0.149 * | 0.067±0.067 ** |
| K357A | 1.000±0.000 | 1.000±0.000 | 1.000±0.000 | 1.000±0.000 | 1.000±0.000 | 0.960±0.040 | 1.000±0.000 | 1.000±0.000 |

**FIG 4** Drug resistance profiles of *E. coli* BW25113 Δ*emrE*Δ*mdfA* harboring CraA variants with mutation of titratable or hydrophilic amino acids lining along the predicted binding pocket. Data shown are mean ± standard error of mean, $n \geq 4$. * represents $P < 0.05$ and ** represents $P < 0.005$ (Student's *t*-test). CLM: chloramphenicol; FLR: florfenicol; THM: thiamphenicol; BEN: benzalkonium; ETB: ethidium bromide; CLH: chlorhexidine; NOR: norfloxacin; MIC: mitomycin C. Lightest green, 0.0–0.2 (severely affected); light green, 0.2–0.5 (moderately affected); green, 0.5–0.75 (mildly affected); dark green, 0.75–1.2 (unaffected); darkest green, >1.2 (overgrowth). The underlying numerical data are available in Dataset S1.

previous results (36), CraA_Y73A and CraA_Y73L substitution rendered CraA inactive (Fig. 5; Fig. S3A). While Y73F maintained wild-type CraA activity, contrasting with the partial activity of MdfA_Y61F against chloramphenicol (Fig. 5; Fig. S3) (36). Together,

| Variants | CLM 2.5–5 µg/mL | FLR 2.5–5 µg/mL | THM 20–50 µg/mL | ETB 0.2–0.4 mg/mL | BEN 15–30 µg/mL | CLH 2.5–5 µg/mL | NOR 0.1–0.5 µg/mL | MIC 1–2.5 µg/mL |
|---|---|---|---|---|---|---|---|---|
| Y42A | 0.875±0.061 | 0.408±0.142 ** | 0.786±0.082 * | 0.083±0.057 ** | 1.000±0.000 | 1.400±0.269 | 0.917±0.053 | 0.000±0.000 |
| Y42F | 0.979±0.021 | 0.892±0.049 | 0.971±0.029 | 0.878±0.065 | 1.000±0.000 | 1.257±0.129 | 0.800±0.122 | 0.964±0.036 |
| Y42L | 0.000±0.000 | 0.222±0.159 ** | 0.000±0.000 | 0.056±0.035 ** | 1.000±0.000 | 1.400±0.269 | 0.200±0.097 ** | 0.845±0.069 * |
| M70A | 0.833±0.000 ** | 0.367±0.122 ** | 1.000±0.000 | 0.778±0.093 * | 0.944±0.056 | 1.000±0.000 | 0.000±0.000 | 0.000±0.000 |
| Y73A | 0.000±0.000 | 0.000±0.000 | 0.000±0.000 | 0.000±0.000 | 0.148±0.076 ** | 0.342±0.114 ** | 0.000±0.000 | 0.000±0.000 |
| Y73F | 1.000±0.000 | 1.000±0.000 | 1.000±0.000 | 1.000±0.000 | 1.050±0.050 | 1.050±0.050 | 1.000±0.000 | 0.833±0.096 |
| Y73L | 0.000±0.000 | 0.000±0.000 | 0.000±0.000 | 0.000±0.000 | 0.000±0.000 | 0.000±0.000 | 0.000±0.000 | 0.000±0.000 |
| L74A | 0.5±0.096 ** | 0.333±0.105 ** | 0.556±0.147 * | 0.167±0.068 ** | 1.000±0.000 | 1.125±0.125 | 0.333±0.096 ** | 0.167±0.136 ** |
| L131A | 0.791±0.042 ** | 0.133±0.062 ** | 0.667±0.096 * | 1.000±0.000 | 0.944±0.056 | 0.083±0.083 ** | 1.000±0.000 | 0.028±0.028 ** |
| Y139A | 1.000±0.000 | 1.000±0.000 | 1.000±0.000 | 1.000±0.000 | 1.000±0.000 | 0.396±0.115 ** | 0.375±0.125 ** | 0.333±0.158 ** |
| M158A | 1.000±0.000 | 0.667±0.139 * | 1.000±0.000 | 0.708±0.197 | 1.000±0.000 | 0.333±0.152 ** | 0.417±0.16 * | 1.000±0.000 |
| L163A | 0.555±0.102 ** | 1.000±0.000 | 0.146±0.080 ** | 1.000±0.000 | 1.071±0.071 | 0.500±0.0960 * | 1.545±0.450 | 1.083±0.083 |
| P166A | 0.083±0.048 ** | 0.1±0.067 ** | 0.111±0.111 ** | 0.208±0.158 ** | 1.000±0.000 | 0.521±0.098 * | 0.625±0.042 | 0.000±0.000 |
| L246A | 0.944±0.056 | 1.000±0.000 | 1.000±0.000 | 1.000±0.000 | 1.000±0.000 | 1.000±0.000 | 1.5±0.224 * | 0.833±0.086 |
| M247A | 0.944±0.056 | 1.000±0.000 | 1.000±0.000 | 0.167±0.096 ** | 1.000±0.000 | 1.000±0.000 | 1.000±0.000 | 0.000±0.000 |
| I250A | 0.000±0.000 | 0.667±0.068 ** | 0.611±0.056 ** | 0.583±0.068 ** | 1.000±0.000 | 1.000±0.000 | 1.000±0.000 | 0.000±0.000 |
| Y268A | 0.444±0.147 * | 0.792±0.125 | 0.611±0.056 ** | 1.000±0.000 | 1.000±0.000 | 1.000±0.000 | 1.000±0.000 | 1.000±0.000 |
| F276A | 1.000±0.000 | 0.708±0.125 | 1.000±0.000 | 1.000±0.000 | 1.000±0.000 | 1.000±0.000 | 0.000±0.000 | 0.5±0.155 * |
| L279A | 0.167±0.096 ** | 0.417±0.048 * | 0.000±0.000 | 1.000±0.000 | 1.000±0.000 | 0.333±0.075 ** | 1.000±0.000 | 0.5±0.155 * |
| F342A | 1.000±0.000 | 0.389±0.035 ** | 0.944±0.035 | 0.944±0.035 | 1.000±0.000 | 1.000±0.000 | 0.833±0.075 | 0.000±0.000 |
| Y346A | 1.000±0.000 | 0.600±0.100 ** | 0.938±0.030 | 0.000±0.000 | 0.833±0.089 | 0.444±0.056 ** | 0.045±0.045 ** | 0.548±0.140 * |
| L368A | 1.000±0.000 | 1.000±0.000 | 1.000±0.000 | 0.972±0.028 | 0.976±0.024 | 0.5±0.096 * | 1.272±0.183 | 1.000±0.000 |
| M369A | 1.000±0.000 | 0.667±0.075 ** | 1.000±0.000 | 1.000±0.000 | 1.000±0.000 | 0.444±0.147 * | 1.106±0.100 | 0.750±0.089 * |
| F372A | 1.000±0.000 | 0.944±0.056 | 1.000±0.000 | 0.083±0.037 ** | 1.1.00±0.100 | 0.54±0.129 * | 0.233±0.194 ** | 0.000±0.000 |

**FIG 5** Drug resistance profiles of *E. coli* BW25113 Δ*emrE*Δ*mdfA* harboring CraA variants with mutation of hydrophobic amino acids lining the predicted binding pocket. Data shown are mean ± standard error of mean, $n \geq 4$. * represents $P < 0.05$ and ** represents $P < 0.005$ (Student's *t*-test). CLM: chloramphenicol; FLR: florfenicol; THM: thiamphenicol; BEN: benzalkonium; ETB: ethidium bromide; CLH: chlorhexidine; NOR: norfloxacin; MIC: mitomycin C. Lightest green, 0.0–0.2 (severely affected); light green, 0.2–0.5 (moderately affected); green, 0.5–0.75 (mildly affected); dark green, 0.75–1.2 (unaffected); darkest green, >1.2 (overgrowth). The underlying numerical data are available in Dataset S1.

our results suggest that the R124-Y73 interaction is likewise essential for the transport function in CraA, comparable to the R112-Y61 interaction in MdfA.

## Residues lining the binding cavity with defined roles in drug transport

Based on our homology model and docking data, we identified 24 residues lining the CraA binding pocket for alanine-scanning mutagenesis (Fig. S8). All *craA* mutants were expressed at comparable levels to the WT when expressed from the same expression vector (Fig. S3B). Most alanine-substitution variants exhibited varying degrees of susceptibility against the tested substrates, underscoring the remarkable promiscuity of the CraA binding pocket in accommodating and transporting structurally diverse compounds (Fig. 4 and 5; Fig. S2 and S3A). Among the residues lining the binding pocket of CraA, N45A, L74A, and N283A variants consistently showed reduced resistance across all tested drugs (Fig. 4 and 5; Fig. S3A). Docking simulations revealed that N45 forms hydrogen bonds with chloramphenicol and norfloxacin (Fig. 1B, 3B, and E), akin to the roles of its homolog N33 in MdfA (Fig. 3A) (36). L74 likely participates in hydrophobic or van der Waals interactions with all substrates (Fig. 3B through E), while N283, though not directly interacting with ligands (Fig. 3), still impairs drug transport when substituted with alanine. These results suggest that N45 and L74 are directly involved in substrate binding, while N283 may play an indirect but functionally significant role in the transport cycle.

Additionally, Y42, homologous to Y30 of MdfA (Fig. S1), also emerged as a key residue in chloramphenicol binding (36). Docking simulations revealed that Y42 of CraA engages in van der Waals interaction with chloramphenicol (Fig. 3B), π-stacking interaction with ethidium, mitomycin C, and norfloxacin (Fig. 3C through E), in addition to forming hydrogen bonding with mitomycin C (Fig. 3D). Phenotypic analysis showed that the Y42A substitution abolished resistance to florfenicol, ethidium, and mitomycin C but retained resistance against chloramphenicol, thiamphenicol, benzalkonium, chlorhexidine, and norfloxacin (Fig. 5; Fig. S3A). Interestingly, substitution of Y42 with leucine impaired resistance to phenicols, while Y42F substitution restored full activity against phenicols (Fig. 5; Fig. S3A), highlighting the critical role of the aromatic side chain at this position for CraA function.

## Rate of ethidium efflux

Ethidium bromide is one of the standard probes for the study of efflux pump activity (12, 40). In order to quantitatively analyze the effect of side chain substitutions in CraA on ethidium efflux, we fitted the ethidium efflux curves of *E. coli* BW25113 Δ*emrE*Δ*mdfA* cells harboring either the empty vector or the plasmid pTTQ18 harboring the *craA* wild-type gene or its mutants (resulting in the expression of the E38A, D46A, E338A, or E38A-D46A variants) (Fig. S9) to a sigmoid function (equation 1) and calculated the $t_{\text{efflux-50\%}}$ from the $H$ parameter, where $H$ is the log of the time required for the cells to remove half of the ethidium. Congruent to the qualitative analysis of the ethidium curve (12), cells harboring an empty vector required an average of ~2,600 s to reach 50% of the ethidium fluorescence, which is a ~16-fold decrease in efflux activity compared to cells harboring wild-type CraA (average of ~160 s) (Table 1; Fig. S9 and S10A). Interestingly, neutralization of D46 only slightly decreased the $t_{\text{efflux-50\%}}$ as compared to wild-type efflux activity (660 s, $P < 0.001$, Tukey's multiple comparison test) (Table 1; Fig. S9 and S10A). As expected, both E38A and double-alanine substituted E38A_D46A mutants showed severely reduced ethidium efflux activity by ~15- and ~59-fold decrease in $t_{\text{efflux-50\%}}$ (an average of ~2,500 and 9,500 s), respectively, compared to wild-type CraA (Table 1; Fig. S9 and S10A). The ethidium efflux activity of cells overproducing the E38A_D46A was slower than the cells harboring an empty vector (an average of $t_{\text{efflux-50\%}}$ = ~9,500 s, $P = <0.001$, Tukey's multiple comparison test) (Table 1; Fig. S9 and S10A). Since the kinetics of ethidium efflux in the E38A_D46A variant were slower than cells harboring the empty vector, we extended our drug susceptibility assays to include lower concentrations of drugs and compared the growth of this variant to cells with the

**TABLE 1** Ethidium and norfloxacin efflux of CraA wild type and variants[a,b]

| Variants | Ethidium efflux | | Norfloxacin accumulation |
| --- | --- | --- | --- |
| | $H$ | $t_{\text{efflux-50\%}}$ | Relative fluorescence |
| Wild-type (WT) | 5.08 ± 0.09 | 160.77 | 0.996 ± 0.08 |
| Vector | 7.86 ± 0.09*** | 2,591.52 | 2.247 ± 0.08*** |
| E38A | 7.81 ± 0.16*** | 2,465.13 | 3.230 ± 0.18*** |
| E38A_D46A | 9.16 ± 0.25*** | 9,509.06[$] | – |
| Y42A | 6.40 ± 0.13*** | 601.85 | 2.180 ± 0.12*** |
| Y42F | 5.89 ± 0.13*** | 361.41 | – |
| Y42L | 6.61 ± 0.13*** | 742.48 | – |
| N45A | 6.60 ± 0.16*** | 735.10 | 3.265 ± 0.14*** |
| D46A | 6.52 ± 0.17*** | 678.58 | 1.575 ± 0.11*** |
| D46N | – | – | 1.400 ± 0.13 |
| Y73A | 6.35 ± 0.12*** | 572.49 | 2.473 ± 0.14*** |
| Y73L | – | – | 2.796 ± 0.20*** |
| Y73F | 5.14 ± 0.15 | 170.72 | 1.090 ± 0.15 |
| L74A | 6.37 ± 0.16*** | 584.06 | – |
| R124A | 7.93 ± 0.29*** | 2,779.43 | – |
| R124H | 7.52 ± 0.29*** | 1,844.57 | – |
| L131A | 4.78 ± 0.16 | 119.10 | – |
| M158A | 5.50 ± 0.16 | 244.69 | – |
| L163A | 4.68 ± 0.14 | 107.77 | 0.692 ± 0.15 |
| L246A | 4.38 ± 0.16*** | 79.84 | 0.689 ± 0.18 |
| M247A | 5.41 ± 0.29 | 223.63 | – |
| I250A | 6.26 ± 0.14*** | 523.22 | 2.085 ± 0.15*** |
| Y268A | 5.44 ± 0.15 | 230.44 | 0.821 ± 0.15 |
| L279A | 4.95 ± 0.14 | 141.17 | 0.445 ± 0.17 |
| N283A | 6.25 ± 0.16*** | 518.01 | 3.109 ± 0.14*** |
| E338A | 5.19 ± 0.11 | 179.47 | 2.528 ± 0.18*** |
| E338D | 5.99 ± 0.11*** | 399.41 | – |
| E338I | 5.18 ± 0.14 | 177.68 | – |
| E338N | 5.35 ± 0.11 | 210.61 | – |
| E338Q | 5.73 ± 0.11*** | 307.97 | – |
| Y346A | 5.53 ± 0.16 | 252.14 | 1.867 ± 0.12*** |
| M369A | 4.55 ± 0.16* | 94.63 | – |
| F372A | 5.48 ± 0.16 | 239.85 | 1.817 ± 0.17*** |

[a]The log of the time required for the cells to remove half of the ethidium, parameter $H$, of each CraA variant was calculated by fitting the curves of ethidium efflux to a sigmoid function as described in Materials and Methods. $t_{\text{efflux-50\%}}$, the time needed to reach a relative fluorescence of 0.5 is calculated by fitting the estimated $H$ to the exponentiation function, $f(H) = e^H$. The relative fluorescence of accumulated norfloxacin in each CraA variant was analyzed by a linear mixed effects model as described in Materials and Methods. Statistical comparisons between the cells harboring WT and CraA variants were performed using Tukey's multiple comparison test, *** represents $P < 0.001$, * represents $P < 0.05$, [$] represents $P < 0.005$ (statistical comparison between the cells harboring E38A_D46A variant and empty vector). All CraA variants were expressed equally well compared to wild-type CraA (Fig. S3B).
[b] – represents not determined.

empty vector. Our results showed that both the E38A and E38A_D46A variants exhibited similar susceptibility to phenicols, ethidium, and norfloxacin as the empty vector control (Fig. S11). However, these variants displayed reduced resistance to benzalkonium and mitomycin C compared to cells with the empty vector (Fig. S11). These findings suggest that alanine substitution at both E38 and D46 impairs the efflux activity of phenicols, ethidium, and norfloxacin, while potentially facilitating the influx of benzalkonium and mitomycin C. In line with the drug susceptibility assays, cells expressing *craA_E338A*, *craA_E338I*, and *craA_E338N* displayed ethidium transport activity to levels comparable to wild-type CraA (Table 1; Fig. 4; Fig. S9 and S10A). However, the E338D variant exhibited a >2.5-fold increase in $t_{\text{efflux-50\%}}$ (~400 s) in ethidium efflux activity compared to wild-type CraA (Table 1; Fig. S9 and S10A). Surprisingly, the E338Q variant displayed a

small but statistically significant increase in $t_{efflux\text{-}50\%}$ compared to CraA wild type (an average of $t_{efflux\text{-}50\%}$ = ~310 s, $P$ = <0.001, Tukey's multiple comparison test) (Table 1; Fig. S9 and S10A). This finding aligns with drug susceptibility assays, which also revealed a small, statistically significant decrease in resistance (Fig. 4; Fig. S3A).

In accord with the drug susceptibility assays, alanine substitution of residues lining the binding site of ethidium (N45, L74, and I250) as predicted from the *in silico* modeling, and residue R124 diminished the efflux activity of ethidium (Table 1; Fig. 4 and 5; Fig. S3A, S9, and S10A). Surprisingly, CraA_M247A, Y346A, and CraA_F372A mutants only displayed a relatively small, statistically insignificant increase in $t_{efflux\text{-}50\%}$ compared to CraA wild type (Table 1; Fig. S9 and S10A), differing from the data obtained from drug susceptibility assays (Fig. 5; Fig. S3A). We reasoned that alanine substitution of M247, Y346, and F372 might not curtail the transport activity of CraA in a relatively lower concentration of ethidium bromide, as only 10 µM (equivalent to ~4 µg mL$^{-1}$) was used in the ethidium efflux assays (refer to Materials and Methods). However, the impact of an increase of $t_{efflux\text{-}50\%}$ in CraA_M247A (~220 s), Y346A (~250 s), and CraA_F372A (~240 s) is likely to be augmented when the concentration of ethidium bromide is considerably increased (300 µg mL$^{-1}$) as shown in the drug susceptibility assays (Fig. 5; Table 1; Fig. S3A, S9, and S10A). Therefore, we anticipated that these variants might render the cells highly susceptible to high levels of ethidium concentration, suggesting a role of M247, Y346, and F372 in the transport of ethidium. In line with the drug susceptibility assays, cells expressing *craA_Y42A* and *craA_Y73A* exhibited a >3-fold increase in $t_{efflux\text{-}50\%}$ (Y42A: ~600 s; Y73A: ~570 s) in ethidium efflux activity compared to wild-type CraA (Fig. 5; Table 1; Fig. S3A, S9, and S10A), suggesting a low efflux activity for this substrate. In contrast, substitution of tyrosine with phenylalanine at position 73 restored ethidium transport activity to levels comparable to wild-type CraA ($t_{efflux\text{-}50\%}$ = 170 s), while the substitution at position 42 partially restored ethidium transport activity ($t_{efflux\text{-}50\%}$ = 360 s) (Table 1; Fig. S9 and 10A).

To further validate the CraA proton-driven mechanism, CCCP was added to the cell suspension 1 min before glucose addition, and fluorescence was monitored for 8 min. As anticipated, the $H$ parameter values for cells harboring wild-type CraA, the empty vector, and the D46A variant in the presence of DMSO were consistent with those obtained from efflux assays performed without CCCP (Table 1; Table S2). Notably, CCCP significantly reduced the ethidium efflux rate in wild-type CraA-expressing cells, resulting in a 3.5-fold increase in $t_{efflux\text{-}50\%}$ (Table S2; Fig. S9B). While CCCP also impaired efflux in cells harboring the D46A variant, the differences were not statistically significant (Table S2; Fig. S9B). Our results confirm that CraA activity is sensitive to disruption of the proton electrochemical gradient.

## Norfloxacin accumulation assays

Since D46A and D46N mutants retained resistance to norfloxacin (Fig. 4; Fig. S3A), we further addressed the norfloxacin accumulation of cells expressing these mutants. To test this notion, we first ascertained the norfloxacin efflux activity of wild-type CraA by determining the accumulation of norfloxacin in *E. coli* BW25113 Δ*emrE*Δ*mdfA* by overexpressing the wild-type *craA* and cells harboring an empty vector (pTTQ18) as a negative control. Indeed, our data clearly showed that a significant amount of norfloxacin was accumulated within the cells harboring pTTQ18 (estimated fluorescence = 2.25 ± 0.08, $P$ < 0.001, Tukey's multiple comparison test) compared to the cells overexpressing wild-type *craA* (estimated fluorescence = 1.0 ± 0.08), as indicated by a high fluorescence intensity of norfloxacin in cells harboring empty vector (Table 1; Fig. S10B). In line with the results of the drug susceptibility assays, the norfloxacin accumulation assays also revealed that the membrane-embedded negatively charged residues (E38 and E338) are important in the transport of norfloxacin, as these mutants CraA_E38A (relative fluorescence = 3.230 ± 0.18, $P$ < 0.001, Tukey's multiple comparison test compared to empty vector) and CraA_E338A (relative fluorescence = 2.528 ± 0.18, $P$ = 0.98, Tukey's multiple comparison test compared to empty vector) displayed

exacerbated transport activity, as these mutants accumulated even more norfloxacin compared to cells harboring pTTQ18 (Table 1; Fig. S10B). Confoundingly, in addition to the E38A and E338A variants, both the N45A and N283A variants exhibited a decrease in efflux activity, in which the relative norfloxacin fluorescence was higher than empty vector (Table 1; Fig. S10B). Noteworthy, the D46A variant conferred some degree of resistance against norfloxacin (relative fluorescence = 1.6 ± 0.11), but to a lesser extent compared to wild type as this mutant accumulated more norfloxacin than WT ($P <$ 0.001, Tukey's multiple comparison test), but less than the negative control ($P < 0.001$, Tukey's multiple comparison test) (Table 1; Fig. S10B). The D46N variant, on the other hand, showed wild-type CraA norfloxacin efflux activity (relative fluorescence = 1.4 ± 0.13, $P$ = 0.15696, Tukey's multiple comparison test) (Table 1; Fig. S10B). As expected, alanine substitution of residues lining the norfloxacin binding site, as predicted from the *in silico* modeling such as Y42, N45, I250, and F372, resulted in an increased level of norfloxacin accumulation within the cells overproducing the CraA variants, except for CraA_L246A and CraA_L279A mutants (Table 1; Fig. 4 and 5; Fig. S10B). Congruent with the drug susceptibility assays, the CraA_L246A and CraA_L279A variants consistently and reproducibly accumulated less norfloxacin compared to that of WT, even though the results are statistically non-significant (Fig. 5; Table 1; Fig. S10B). Since Y73 is highly sensitive to amino acid substitution (Fig. 5; Fig. S3A), we measured the accumulation of norfloxacin in cells expressing *craA_Y73A*, *craA_Y73L*, and *craA_Y73F* mutants. As expected, a relatively high norfloxacin fluorescence was measured in CraA_Y73A and CraA_Y73L mutants (estimated fluorescence: CraA_Y73A = 2.5 ± 0.14; Y73L = 2.8 ± 0.20; $P < 0.001$, Tukey's multiple comparison test) (Table 1; Fig. S10B). In contrast, substitution of Y73 to phenylalanine displayed wild-type norfloxacin efflux activity (estimated fluorescence = 1.1 ± 0.15) (Table 1; Fig. S10B).

## DISCUSSION

Chloramphenicol is rarely used in humans today due to its severe hematological side effects, but it remains an alternative for life-threatening infections like meningitis, septicemia, and typhoid fever (41, 42). It is still widely used in animal farming and aquaculture in some developing countries (43) and is frequently detected in aquatic environments worldwide (44). In this study, we investigate the polyspecificity of an efflux pump initially characterized as highly specific for chloramphenicol (12, 15). While the regulation of the *craA* gene expression is not fully understood, we observed that *craA* is not constitutively expressed ($\Delta C_T$ = −4.7 ± 0.3) but is highly responsive to chloramphenicol ($\Delta\Delta C_T$ = 4.0 ± 0.5, fold change >16, $P < 0.001$) (Fig. S12). Therefore, the presence of chloramphenicol in the environment, along with exposure of clinical and environmental isolates, should be closely monitored to prevent overexpression of this gene.

Docking simulations reveal that E338 is positioned distally relative to ethidium and chloramphenicol (Fig. 6C). However, it forms a hydrogen bond with the urethane group of mitomycin C and is in close proximity to norfloxacin (Fig. 6D and E). $pK_a$ calculations using PROPKA3.2 (45) predict that E338 is protonated under physiological conditions, with a pKa of ~8.3 (Table S3). For comparison, the $pK_a$ values of E38 and D46 in CraA and their homologous residues E26 and D34 in the MdfA crystal structure (PDB: 4ZOW) are similar, supporting the reliability of the predictions (Table S3). Interestingly, in the presence of norfloxacin, the $pK_a$ of E338 drops to ~6.1, indicating that E338 may be deprotonated during transport. Given that norfloxacin $pK_a$ for the piperazinyl ring is ~8.5 (46), we speculate that the negatively charged E338 side chain stabilizes the protonated piperazinyl group of norfloxacin during the transport cycle (Fig. 6E). However, a protonated mimic at this position (E338N and E338Q) cannot interact effectively with protonated piperazine. Interestingly, both E338N and E338Q substitutions restore CraA activity against mitomycin C and phenicols (Fig. 4; Fig. S3A). Our analysis shows that E338 is protonated in docking poses for chloramphenicol and ethidium (Table S3; Fig. 6B and C). In contrast, the $pK_a$ for E338 in the mitomycin C docking pose is ~7.1 (Table S3), suggesting a weaker acid. We propose that the protonated glutamate side chain

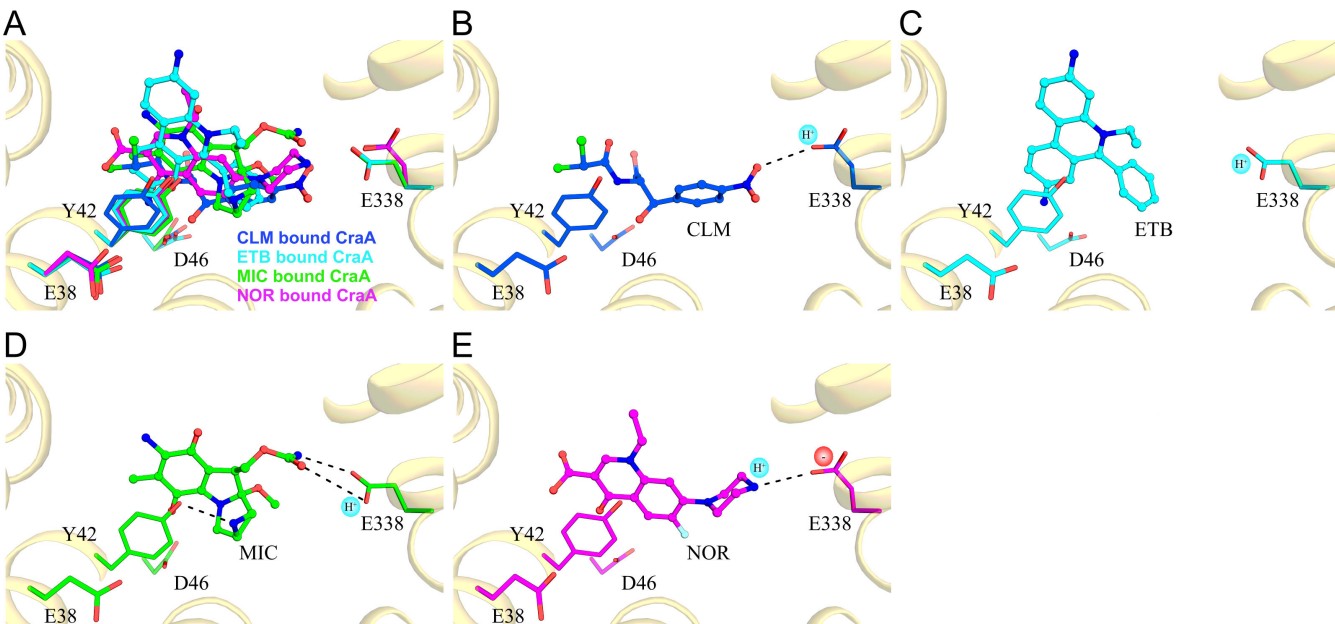

**FIG 6** Y42 and E338 as selective determinants for substrate specificity in CraA. (A) Superimposition of substrate docking solutions for chloramphenicol (carbon: blue), ethidium (carbon: cyan), mitomycin C (carbon: green), and norfloxacin (carbon: magenta). Substrate docking solutions for (B) chloramphenicol (CLM); (C) ethidium (ETB); (D) mitomycin C (MIC); and (E) norfloxacin (NOR), in the CraA binding pocket. Glu-338 is predicted to be protonated in docking solutions for chloramphenicol, ethidium, and mitomycin C, while it is deprotonated in the docking solution for norfloxacin. Carbon: blue/cyan/green/magenta; nitrogen: dark blue; oxygen: red; chlorine: green.

facilitates interactions with mitomycin C and phenicols (Fig. 6). Replacing E338 with a carboxamide likely mimics this protonated state. Overall, our data suggest that E338 is involved in drug selectivity during the transport cycle, though it is not essential for energization.

Florfenicol differs from other phenicols by having a fluorine atom at C4 instead of a hydroxyl group (Fig. S2), which likely disrupts hydrogen bonding with N45 (Fig. 3B). This leads to a distinct binding orientation where Y42 forms a stronger interaction with the dichloroacetamide moiety of florfenicol compared to other phenicols. Whereas the bulky, non-planar aliphatic side chain of leucine at position 42 disrupts the binding of phenicols compared to the Y42A variant (Fig. 5; Fig. S3A), suggesting that a planar phenol or phenyl group is essential for the selectivity of phenicols. Intriguingly, the Y42L substitution restores mitomycin C efflux but reduces resistance to norfloxacin (Fig. 5; Fig. S3A), implying that a less bulky hydrophobic side chain at position 42 is critical for mitomycin C selectivity but not for norfloxacin. Overall, Y42 in CraA plays a key role in substrate specificity through its aromaticity and hydrophobic interactions.

In conclusion, our studies involving structural modeling, *in silico* prediction of substrate binding, site-directed mutagenesis, drug susceptibility tests, and drug transport assays revealed that structural similarity and overlapping substrate specificities between homologous proteins do not always infer a similar transport mechanism; rather, there are species-specific differences among homologs or orthologues. We hypothesize that the outer membrane impermeability of *A. baumannii* compared to *E. coli* and/or the involvement of other transporters may account for the observed differences in drug binding and/or transport between CraA and MdfA (47, 48). CraA exhibits a certain degree of similarity with *E. coli* MdfA (like E38, N45, and D46), but with fundamental differences in drug recognition, in which Y42 and E338 in CraA are key substrate selectivity-determining residues, as well as the role of R124 and Y73 in maintaining the transport activity in CraA. Further studies, including high-resolution structures of CraA in complex with bound substrates, are needed to unravel the transport mechanism of CraA.

## ACKNOWLEDGMENTS

We thank Jochen Wilhelm from the Excellence Cluster Cardio-Pulmonary Institute, Justus Liebig University Giessen, for his valuable personal correspondence regarding statistical analysis.

This work was supported by grants from the Deutsche Forschungsgemeinschaft through DFG Research Unit FOR 2251, the start-up package through the University of South China (221RGC012), the Natural Science Foundation of Hunan Province (2024JJ5326), and the Science and Technology Innovation Program of Hunan Province (2024RC9011).

## AUTHOR AFFILIATIONS

[1]Department of Biochemistry and Molecular Biology, Hengyang Medical School, University of South China, Hengyang, Hunan, China

[2]Department of Medical Microbiology, Hunan Provincial Key Laboratory for Special Pathogens Prevention and Control, Hengyang Medical School, University of South China, Hengyang, Hunan, China

[3]Institute of Biochemistry, Goethe‐University Frankfurt, Frankfurt am Main, Germany

[4]National Health Commission Key Laboratory of Birth Defect Research and Prevention, Hunan Provincial Maternal and Child Health Care Hospital, Changsha, Hunan, China

## AUTHOR ORCIDs

Wuen Ee Foong http://orcid.org/0000-0003-2402-7729
Klaas M. Pos http://orcid.org/0000-0001-9035-3827
Heng-Keat Tam http://orcid.org/0000-0002-7894-1957

## FUNDING

| Funder | Grant(s) | Author(s) |
|---|---|---|
| Deutsche Forschungsgemeinschaft | DFG Research Unit FOR 2251 | Klaas M. Pos |
| University of South China | 221RGC012 | Heng-Keat Tam |
| HSTD \| Natural Science Foundation of Hunan Province (湖南省自然科学基金) | 2024JJ5326 | Heng-Keat Tam |
| Science and Technology Innovation Program of Hunan Province | 2024RC9011 | Heng-Keat Tam |
| | | Wuen Ee Foong |

## AUTHOR CONTRIBUTIONS

Wuen Ee Foong, Formal analysis, Funding acquisition, Investigation, Methodology, Supervision, Validation, Visualization, Writing – original draft, Writing – review and editing | Xinxin Xiang, Investigation, Validation, Visualization | Klaas M. Pos, Conceptualization, Funding acquisition, Project administration, Supervision, Writing – review and editing | Heng-Keat Tam, Conceptualization, Formal analysis, Funding acquisition, Investigation, Methodology, Project administration, Supervision, Visualization, Writing – original draft, Writing – review and editing

## ADDITIONAL FILES

The following material is available online.

### Supplemental Material

**Data Set S1 (Spectrum01119-25-s0001.xlsx).** Underlying data for Fig. 5 and 6.
**Supplemental material (Spectrum01119-25-s0002.pdf).** Tables S1 to S3; Fig. S1 to S12.

Open Peer Review

**PEER REVIEW HISTORY (review-history.pdf).** An accounting of the reviewer comments and feedback.

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
