## [Reviewer comments · Microbiology Spectrum]

Microbiology Spectrum

Molecular determinants of substrate specificity in CraA efflux pump of *Acinetobacter baumannii*

Wuen Ee Foong, Xinxin Xiang, Klaas Pos, and Heng-Keat Tam

Corresponding Author(s): Heng-Keat Tam, University of South China Hengyang Medical School

Review Timeline:

Submission Date:	April 12, 2025
Editorial Decision:	May 5, 2025
Revision Received:	May 24, 2025
Accepted:	June 8, 2025

Editor: Paolo Visca

Reviewer(s): Disclosure of reviewer identity is with reference to reviewer comments included in decision letter(s). The following individuals involved in review of your submission have agreed to reveal their identity: Lan Guan (Reviewer #2)

Transaction Report:

DOI: <https://doi.org/10.1128/spectrum.01119-25>

Re: Spectrum01119-25 (Molecular determinants of substrate specificity in CraA efflux pump of *Acinetobacter baumannii*)

Dear Dr. Heng-Keat Tam:

Thank you for the privilege of reviewing your work. Below you will find my comments, instructions from the Spectrum editorial office, and the reviewer comments.

Your work provides interesting information on the *A. baumannii* CraA multidrug MFS transporter and is well suited for Spectrum. It has been reviewed by two experts who recommended major revisions.

Please, address all the reviewers' comments and modify the manuscript accordingly, then return the manuscript within 60 days; if you cannot complete the modification within this period, please contact me. If you do not wish to modify the manuscript and prefer to submit it to another journal, notify me immediately so that the manuscript may be formally withdrawn from consideration by Spectrum.

Revision Guidelines

Sincerely,
Paolo Visca
Editor
Microbiology Spectrum

Reviewer #1 (Comments for the Author):

This manuscript describes a thorough characterization of the *A. baumannii* CraA multidrug MFS transporter in comparison with the ortholog *E. coli* transporter MdfA. The role in ligand recognition and transport of several residues lining the CraA ligand

binding cavity has been studied through site directed mutagenesis, drug susceptibility assays, ethidium efflux assays and norfloxacin accumulation assays. Experiment were guided by building a CraA structural model used to dock various drugs into the ligand binding cavity.

A significant amount of data has been produced and the conclusions of the study are largely supported by the results presented. However, the flow of the manuscript is very difficult to follow, especially as far as the results section (mostly pages 13 to 18 of the manuscript) is concerned. Figure 3 doesn't help so much in this regard as it is very difficult to discern the molecular details of the interactions established by cavity-lining residues with the various drugs. A zoom in on the binding site region would probably do a better job.

Adding to the confusion, a couple of sentences in this section are unclear and probably the result of typos.

For instance, on lines 311-312 it is stated that "...carboxylate retaining variants such as E38D_D46A, E38A_D46E, and E38D_D46E did not confer any resistance to all tested drugs." But in the following sentence it is said that "One surprising exception was wildtype like resistance mediated by the exchange variant E38D_D46E, which retained wildtype activity against chlorhexidine (Table 1, Fig. S3A)". Thus, the reader does not understand which is the real effect of the mutation E38D_D46E. Has it a wild type like resistance to all drugs or only against chlorhexidine?

Another example (lines 368-370) "Whereas cells harbouring CraA_Y73A variant were severely susceptible to all the tested drugs (Table 2, Fig. S3A), cells containing Y73L did not show any resistance phenotype against any of the tested drugs". To me this means that both variants are susceptible to all drugs but they are presented like variants that cause different and contrasting effects.

I suggest that the authors make their best effort to improve this section to make it more reader friendly especially for those who are not really into the specific topic of the manuscript.

Further, I may have missed some point but I do not agree with the conclusion that the data presented demonstrate a role of R124 and Y73 in maintaining the structural integrity of CraA. For sure they are very important for the transport activity but more solid evidence would be needed to claim a role in the stability of the transporter.

Minor points

Consider revising sentences like "compute the calculation", lines 265 and 268;

eliminate "was superimposed" from line 290;

Correct the sentence "We reasoned that alanine substituted of M247, Y346A, and F372..." into "We reasoned that alanine substitution of M247, Y346, and F372..."

Line 473, change Y346A into Y346

Reviewer #2 (Comments for the Author):

This manuscript, "Molecular determinants of substrate specificity in CraA efflux pump of *Acinetobacter baumannii*," focuses on the H⁺-coupled MFS multidrug efflux antiporter CraA of *Acinetobacter baumannii*. The multidrug efflux pumps are one of the essential contributors to multidrug resistance in *A. baumannii*, an opportunistic pathogen responsible for nosocomial infections with multidrug resistance. Since the high-resolution structure is not available, the authors generated a model based on the homologue of the *E. coli* MfdA with 44% identity and 64% similarity, and this model is quite similar to the AlphaFold 2 predicted model. Ligand docking and mutagenesis were used to characterize the major side chains in binding and the potential protonation. A classic drug sensitivity assay, plus measurement of transport rates with intact cells. Overall, the experiment designs are sound, and the explanations of results are reasonable.

Comments and suggestions:

1. Page 18, Line 422, the subtitle of "statistical analysis of ethidium efflux". Since all analyses contain statistics, this title can be "rate of ethidium efflux". In the legend to Fig S9, "The comparison of H parameter among different CraA variants is statistically significant with $F(31, 258.29) = 339.57, p < 0.001$ ". It is not clear what this means. A paired comparison between the WT and mutants is still needed, and this information should be included in Table 3.
2. Rate of efflux should be protein concentration-dependent. In the method, a western blot was described to analyze the protein expression, but no western blot results were presented. Since it is a cell-based assay, many other factors could contribute to the efflux rates.
3. Have the authors tested the uncoupler CCCP effects on the rate of ethidium efflux? I understand that CCCP was used for loading and then washed out, which is very well designed. It would be interesting to see the CCCP effects by adding it during the tracing for selected mutants, like D46A.

We have addressed the comments from the reviewers and the answers given are in blue letter-color (please refer also to the tracked changes documents):

Reviewer #1 (Comments for the Author):

This manuscript describes a thorough characterization of the *A. baumannii* CraA multidrug MFS transporter in comparison with the ortholog *E. coli* transporter MdfA. The role in ligand recognition and transport of several residues lining the CraA ligand binding cavity has been studied through site directed mutagenesis, drug susceptibility assays, ethidium efflux assays and norfloxacin accumulation assays. Experiment were guided by building a CraA structural model used to dock various drugs into the ligand binding cavity.

We like to thank the reviewer for the in-depth review of the manuscript.

A significant amount of data has been produced and the conclusions of the study are largely supported by the results presented. However, the flow of the manuscript is very difficult to follow, especially as far as the results section (mostly pages 13 to 18 of the manuscript) is concerned.

We apologize for our lack of clarity and thank you for the valuable suggestion. We have now revised the results section (pages 13 to 18) and please refer to the detailed results (pages 13 to 16 of the clean version of the manuscript).

Figure 3 doesn't help so much in this regard as it is very difficult to discern the molecular details of the interactions established by cavity-lining residues with the various drugs. A zoom in on the binding site region would probably do a better job.

Thank you for the valuable suggestion. We have revised Figure 3 to provide a zoomed-in view of the binding site, highlighting the key residues discussed in the manuscript. However, we believe the original version remains informative in demonstrating the phenotypic effects of surrounding residues on substrate recognition as revealed by alanine-scanning mutagenesis. Therefore, we have moved the original Figure 3 to Supplementary Fig. S5.

Adding to the confusion, a couple of sentences in this section are unclear and probably the result of typos.

We apologize for our lack of clarity and please refer to the answer as the following.

For instance, on lines 311-312 it is stated that "...carboxylate retaining variants such as E38D_D46A, E38A_D46E, and E38D_D46E did not confer any resistance to all tested drugs." But in the following sentence it is said that "One surprising exception was wildtype like resistance mediated by the exchange variant E38D_D46E, which retained wildtype activity against chlorhexidine (Table 1, Fig. S3A)". Thus, the reader does not understand which is the real effect of the mutation E38D_D46E. Has it a wild type like resistance to all drugs or only against chlorhexidine?

We apologize for our lack of clarity and thank you for the valuable suggestion.

We have amended the following sentence on Page 13, Line 301 of the clean version of the manuscript:

“Notably, the CraA double substitution variants E38A_D46A, E38D_D46A, and E38A_D46E were inactive against all tested drugs, while the E38D_D46E variant retained wildtype activity specifically against chlorhexidine (Table 1, Fig. S3A).”

Another example (lines 368-370) "Whereas cells harbouring CraA_Y73A variant were severely susceptible to all the tested drugs (Table 2, Fig. S3A), cells containing Y73L did not show any resistance phenotype against any of the tested drugs". To me this means that both variants are susceptible to all drugs but they are presented like variants that cause different and contrasting effects.

We apologize for our lack of clarity.

We have amended the following sentence on Page 15, Line 345 of the clean version of the manuscript:

“ In line with previous results (Heng 2015), CraA_Y73A and CraA_Y73L substitution rendered CraA inactive (Table 2, Fig. S3A).”

I suggest that the authors make their best effort to improve this section to make it more reader friendly especially for those who are not really into the specific topic of the manuscript.

We apologize for our lack of clarity. Please refer to the answer above.

Further, I may have missed some point but I do not agree with the conclusion that the data presented demonstrate a role of R124 and Y73 in maintaining the structural integrity of CraA. For sure they are very important for the transport activity but more solid evidence would be needed to claim a role in the stability of the transporter.

We appreciate the valuable suggestion. We agree with the reviewer that R124 and Y73 might not play a role in maintaining the structural integrity of CraA, but are essential in transport activity. According to Heng et al. and Nagarathinam et al., the corresponding R112 in MdfA (R124 in CraA) within the R¹¹²xxQG motif participates in a hydrogen network with surrounding residues and cation- π interaction with Y61 (Y73 in CraA) (refer to Fig. S7), suggesting an essential role in coupling the protonation with substrate binding in MdfA. Based on these observation, we think that in addition to the coupling, the interaction between R124 and Y73 might play a role in conformational changes between inward open and outward open as supported by the structural analysis of MdfA (Please refer to Fig. S7).

We have amended the following sentence on Page 3, Line 53 fo the clean version of the manuscript:

“..... while R124 and Y73 contribute to its transport activity.”

We have amended the following sentence on Page 15, Line 348 of the clean version of the manuscript:

“Together, our results suggest that the R124-Y73 interaction is likewise essential for the transport function in CraA comparable to the R112-Y61 interaction in MdfA.”

References:

Heng J, Zhao Y, Liu M et al. Substrate-bound structure of the *E. coli* multidrug resistance transporter MdfA. *Cell Res* 2015; 25: 1060 – 73.

Nagarathinam K, Nakada-Nakura Y, Parthier C, Terada T, Juge N, Jaenecke F, Liu K, Hotta Y, Miyaji T, Omote H, Iwata S, Nomura N, Stubbs MT, Tanabe M. Outward open conformation of a Major Facilitator Superfamily multidrug/H⁺ antiporter provides insights into switching mechanism. *Nat Commun* 2018; 9: 4005.

Minor points

Consider revising sentences like "compute the calculation", lines 265 and 268;

We apologize for this mistake. We have revised the following sentence on Page 12, Line 259 of the clean version of the manuscript:

For the first docking calculations, the search of poses was performed within a large cubic volume of 46 x 38 x 30 Å³ covering the whole cavity of the protein, with an exhaustiveness of 20 (default 8). After identification of putative binding pockets within the cavity of the protein, the search for docking poses was performed within a cubic volume of 24 x 24 x 24 Å³ and centered at the cavity of the protein (x = 28.662 Å; y = 3.413 Å; z = 23.172 Å), with an exhaustiveness of 120.

eliminate "was superimposed" from line 290;

We apologize for this mistake. We have amended the following sentence on Page 13, Line 284 of the clean version of the manuscript:

“Superimposition of the CraA-chloramphenicol docking model with MdfA-chloramphenicol crystal structure demonstrated.....”

Correct the sentence "We reasoned that alanine substituted of M247, Y346A, and F372..." into "We reasoned that alanine substitution of M247, Y346, and F372..."

We apologize for this mistake. We have amended the following sentence on Page 18, Line 423 of the clean version of the manuscript:

“We reasoned that alanine substitution of M247, Y346, and”

Line 473, change Y346A into Y346

We apologize for this mistake. We have amended the following sentence on Page 18, Line 431 of the clean version of the manuscript:

“.... suggesting a role of M247, Y346, and F372”

Reviewer #2 (Comments for the Author):

This manuscript, "Molecular determinants of substrate specificity in CraA efflux pump of *Acinetobacter baumannii*," focuses on the H⁺-coupled MFS multidrug efflux antiporter CraA of *Acinetobacter baumannii*. The multidrug efflux pumps are one of the essential contributors to multidrug resistance in *A. baumannii*, an opportunistic pathogen responsible for nosocomial infections with multidrug resistance. Since the high-resolution structure is not available, the authors generated a model based on the homologue of the *E. coli* MfdA with 44% identity and 64% similarity, and this model is quite similar to the AlphaFold 2 predicted model. Ligand docking and mutagenesis were used to characterize the major side chains in binding and the potential protonation. A classic drug sensitivity assay, plus measurement of transport rates with intact cells. Overall, the experiment designs are sound, and the explanations of results are reasonable.

We like to thank the reviewer for the insightful and valuable review.

Comments and suggestions:

1. Page 18, Line 422, the subtitle of "statistical analysis of ethidium efflux". Since all analyses contain statistics, this title can be "rate of ethidium efflux".

Thank you for the suggestion. We have revised the subtitle as the following:

Page 17: "Rate of ethidium efflux"

In the legend to Fig S9, "The comparison of H parameter among different CraA variants is statistically significant with $F(31, 258.29) = 339.57, p < 0.001$ ". It is not clear what this means.

We apologize for our lack of clarity, and we appreciate the valuable suggestion. We have incorporated a statement to explain the meaning of this sentence as the following on Page 27 of the Supplementary Materials (previously Fig. S9, revised as Fig. S10):

A Type III analysis of variance using Satterthwaite's method was performed on the linear mixed effects model (see Equation 2 in Materials and Methods) to assess the effect of genotype on the *H* parameter. The analysis revealed a statistically significant difference among CraA variants, with $F(31, 258.29) = 339.57, p < 0.001$.

Similarly, we have also incorporated a statement to explain the meaning of this sentence "The comparison of the estimates of relative fluorescence among different CraA variants is statistically significant with $F(19, 121.26) = 87.61, p < 0.001$." in Fig S9B.

A Type III analysis of variance using Satterthwaite's method was performed on the linear mixed effects model (see Equation 3 in Materials and Methods) to assess the effect of genotype on norfloxacin accumulation as measured by the fluorescence intensity. The analysis revealed a statistically significant difference among CraA variants, with $F(19, 121.26) = 87.61, p < 0.001$.

A paired comparison between the WT and mutants is still needed, and this information should be included in Table 3.

Thank you for this very good suggestion. We have incorporated the paired comparison between the WT and mutants in Table 3 and with a statement as the following in Table 3:

Page 34: Table 3 Ethidium and norfloxacin Statistical comparisons between the cells harbouring WT and CraA variants were performed using Tukey's multiple comparison test, *** represents $p < 0.001$, * represents $p < 0.05$.

2. Rate of efflux should be protein concentration-dependent. In the method, a western blot was described to analyze the protein expression, but no western blot results were presented.

We apologize for our lack of clarity and agree with the reviewer that the efflux activity may correlate with the protein amounts. Our results indicate that all variants were produced at levels comparable to the wildtype, and therefore, densitometric correction would likely not significantly affect the outcome.

Please refer to Fig S3B in Supplementary Materials for the western blot results.

Since it is a cell-based assay, many other factors could contribute to the efflux rates.

We agree with the reviewer that whole-cell assays may be influenced by multiple cellular factors, including membrane permeability, expression levels, membrane potential, and the presence of competing transport systems. To minimize these confounding variables, we ensured consistent expression of all CraA variants from the same vector under identical conditions (Fig. S3B). Furthermore, we complemented the efflux assays with drug susceptibility assays of site-directed CraA variants (Table 1, Table 2, Fig. S3A) and molecular docking analysis, to strengthen the link between observed efflux activity and specific residues involved in substrate recognition and transport. While we recognize that reconstituted or in vitro systems offer more controlled environments, whole-cell assays remain a physiologically relevant approach to assess functional efflux, especially when supported by orthogonal validation strategies as performed here.

We have incorporated a statement to the legend of Table 3 as the following:

“All CraA variants were expressed equally well compared to wildtype CraA (Fig. S3B).”

3. Have the authors tested the uncoupler CCCP effects on the rate of ethidium efflux? I understand that CCCP was used for loading and then washed out, which is very well designed. It would be interesting to see the CCCP effects by adding it during the tracing for selected mutants, like D46A.

Thank you for this very good suggestion. We have performed and incorporated the experiments as following:

Page 8, Line 167 of the clean version of the manuscript:

“To evaluate the effect of CCCP on ethidium efflux, CCCP (40 μ M) was added to the cell suspension and fluorescence was monitored for one minute. Subsequently, glucose (0.36%) was added and fluorescence measurement was continued for an additional 8 min.

Each of the ethidium efflux curve following glucose addition was individually analyzed”

Page 19, Line 439 of the clean version of the manuscript:

“To further validate CraA proton-driven mechanism, CCCP was added to the cell suspension one minute before glucose addition, and fluorescence was monitored for 8 min. As anticipated, the *H* parameter values for cells harbouring wildtype CraA, the empty vector, and the D46A variant in the presence of DMSO were consistent with those obtained from efflux assays performed without CCCP (Table 3, Table S2). Notably, CCCP significantly reduced the ethidium efflux rate in wildtype CraA-expressing cells, resulting in a 3.5-fold increase in $t_{\text{efflux-50\%}}$ (Table S2, Fig. S9B). While CCCP also impaired efflux in cells harboring the D46A variant, the differences were not statistically significant (Table S2, Fig. S9B). Our results confirm that CraA activity is sensitive to disruption of the proton gradient.”

Please refer to the Supplementary Materials (Table S2 and Fig. S9B) for detailed results.

Re: Spectrum01119-25R1 (Molecular determinants of substrate specificity in CraA efflux pump of *Acinetobacter baumannii*)

Dear Dr. Heng-Keat Tam:

Your manuscript has been accepted, and I am forwarding it to the ASM production staff for publication. Your paper will first be checked to make sure all elements meet the technical requirements. ASM staff will contact you if anything needs to be revised before copyediting and production can begin. Otherwise, you will be notified when your proofs are ready to be viewed.

Sincerely,
Paolo Visca
Editor
Microbiology Spectrum

Reviewer #1 (Comments for the Author):

The authors have addressed my concerns and therefore I recommend publication

Reviewer #2 (Comments for the Author):

The authors made significant revisions and included the necessary data. My concerns and comments have been fully addressed.

Minor:
Change proton gradient to proton electrochemical gradient.